# C-type natriuretic peptide/cGMP/FoxO3 signaling attenuates hyperproliferation of pericytes from patients with pulmonary arterial hypertension
Swati Dabral [1] ✉, Minhee Noh[1], Franziska Werner[1], Lisa Krebes[1], Katharina Völker[1], Christopher Maier[2], Ivan Aleksic [2], Tatyana Novoyatleva [3,4,5,6], Stefan Hadzic [3,4,5,6], Ralph Theo Schermuly [3,4,5,6], Vinicio A. de Jesus Perez [7] & Michaela Kuhn[1]

Pericyte dysfunction, with excessive migration, hyperproliferation, and differentiation into smooth muscle-like cells contributes to vascular remodeling in Pulmonary Arterial Hypertension (PAH). Augmented expression and action of growth factors trigger these pathological changes. Endogenous factors opposing such alterations are barely known. Here, we examine whether and how the endothelial hormone C-type natriuretic peptide (CNP), signaling through the cyclic guanosine monophosphate (cGMP) -producing guanylyl cyclase B (GC-B) receptor, attenuates the pericyte dysfunction observed in PAH. The results demonstrate that CNP/GC-B/cGMP signaling is preserved in lung pericytes from patients with PAH and prevents their growth factor-induced proliferation, migration, and transdifferentiation. The anti-proliferative effect of CNP is mediated by cGMP-dependent protein kinase I and inhibition of the Phosphoinositide 3-kinase (PI3K)/AKT pathway, ultimately leading to the nuclear stabilization and activation of the Forkhead Box O 3 (FoxO3) transcription factor. Augmentation of the CNP/GC-B/cGMP/FoxO3 signaling pathway might be a target for novel therapeutics in the field of PAH.

Pulmonary arterial hypertension (PAH) is a rare but severe disease characterized by elevated pulmonary arterial blood pressure, which may provoke progressive right heart failure and death[1]. Pulmonary vascular remodeling, with a switch of pulmonary arterial smooth muscle cells (PASMCs) to a hyperproliferative and hypercontractile phenotype, underlies the pathogenesis of PAH[2,3]. Treatment options are limited, partly due to the incomplete understanding of the cellular mechanisms driving vascular remodeling. So far, most research has been directed toward studying and addressing the changes in the phenotype of PASMCs[4,5]. However, recent studies have demonstrated that pericytes contribute to microvascular remodeling in PAH[6–10]. Hence, understanding the mechanisms of pericyte dysfunction in PAH may help to identify novel targets for therapies.

Pericytes are mural cells surrounding endothelial cells (ECs) in pre-capillary arterioles, capillaries, and postcapillary venules. Apart from regulating vascular tone, pericytes play a crucial role in the synthesis of the capillary basement membrane, establishment of the microvascular barrier and immune surveillance. Moreover, the mutual communication between pericytes and EC maintains microvascular homeostasis[3]. Studies of lungs from patients with PAH revealed that pericytes were detached from the ECs, thereby contributing to small vessel loss[10]. This was attributed to reduced endothelial Wnt5a ligand expression, leading to impaired pericyte recruitment[10]. Moreover, in experimental and human PAH, pericytes cluster in high numbers around distal arteries, revealing their contribution to microvascular remodeling[6]. Concordantly, pericytes isolated from the lungs of PAH patients had augmented proliferation and migration rates, as

[1]Institute of Physiology, University of Würzburg, Würzburg, Germany. [2]Department of Thoracic and Cardiovascular Surgery, University hospital Würzburg, Würzburg, Germany. [3]Justus-Liebig-University Giessen (JLU), Giessen, Germany. [4]Universities of Giessen and Marburg Lung Center (UGMLC), Giessen, Germany. [5]Excellence Cluster Cardio-Pulmonary Institute (CPI), Giessen, Germany. [6]Member of the German Center for Lung Research (DZL), Giessen, Germany. [7]Divisions of Pulmonary and Critical Care Medicine and Stanford Cardiovascular Institute, Stanford University, California, USA. ✉e-mail: swati.dabral@uni-wuerzburg.de

well as enhanced potential to differentiate into α-smooth muscle actin (α-SMA) positive, hypercontractile SMC-like cells. Mechanistically, these alterations, which were observed ex vivo, were attributed to their enhanced exposure (in situ) to paracrine endothelial growth factors such as Platelet-derived growth factor-BB (PDGF-BB), Interleukin-6 and transforming growth factor beta (TGF-β). Moreover, such cultured pericytes from patients with PAH had increased expression of receptors for TGF-β and for chemokines (e.g. CXCL) as well as altered metabolism[7,8]. While several factors promoting pericyte dysfunction have been described, endogenous counter-regulators are barely known.

C-type natriuretic peptide (CNP) is a member of the family of natriuretic peptides, which play a crucial role in the regulation of blood pressure. In contrast to the endocrine cardiac hormones Atrial (ANP) and B-type natriuretic peptide (BNP), endothelial CNP acts in an autocrine/paracrine manner. Through its transmembrane guanylyl cyclase-B (GC-B) receptor, forming cyclic GMP as a second messenger, CNP strengthens endothelial barrier integrity, reduces the reactivity of leukocytes and platelets, and prevents atherogenesis[11,12]. In addition, previous studies of our group showed that CNP-induced GC-B/cGMP signaling in microvascular SMCs and pericytes is essential for the maintenance of physiological peripheral microvascular resistance and systemic arterial blood pressure[13]. However, whether CNP also acts on the pulmonary vasculature is barely known. In rats, administration of exogenous synthetic CNP and virus-mediated pulmonary overexpression of a constitutively active GC-B receptor blunted acute hypoxic pulmonary vasoconstriction and monocrotaline-induced lung vessel remodeling, inflammation, and pulmonary hypertension (PH)[14,15]. The cells and precise mechanisms underlying these effects are unclear.

Here, we studied whether CNP/GC-B signaling may modulate the functional and molecular alterations of lung pericytes in patients with PAH. To this aim, we performed in vitro studies with cultured pericytes isolated from human control and PAH lungs. The results demonstrate that CNP/GC-B/cGMP signaling is preserved in PAH pericytes and prevents their growth factor-induced proliferation, migration and transdifferentiation. The antiproliferative effect of CNP is mediated by cyclic GMP-dependent kinase I (cGKI) and inhibition of the phosphatidylinositol 3-kinase (PI3K)/protein kinase B (PKB/AKT) pathway, ultimately leading to stabilization of the Forkhead Box O 3 (FoxO3) transcription factor. Augmentation of the CNP/GC-B/FoxO3 signaling pathway in lung pericytes might be a target for novel therapeutics in the field of PAH.

## Results
### Unaltered CNP/GC-B/cGMP signaling in pericytes from patients with PAH
To compare CNP/cGMP signaling between pulmonary vascular cell types, we performed experiments with cultured human PASMCs as well as with human lung microvascular ECs and pericytes. CNP did not modulate intracellular cGMP levels of ECs, while ANP provoked a small, ≈2-fold increase (Fig. 1a). In contrast, CNP markedly and concentration-dependently increased cGMP levels in PASMCs and, even more, in pericytes (Fig. 1a). As shown, in both types of cells the effects of the highest CNP concentration (100 nM) were much greater than the effect of ANP, used at the same concentration.

Notably, pericytes isolated from the lungs of PAH patients exhibited similar cGMP responses to CNP as the pericytes from control lungs (Fig. 1b). Concordantly, the expression levels of the GC-B receptor and of its downstream target cGKI were similar in control and PAH pericytes (immunoblots in Fig. 1c). In both groups, CNP led to cGKI-mediated phosphorylation of the cytoskeleton-associated vasodilator-stimulated phosphoprotein (VASP) at $Ser_{239}$ (Fig. 1d). Together these results demonstrate that among lung vascular cell types, pericytes exhibit the highest responsiveness to CNP. Moreover, CNP/GC-B/cGMP/cGKI signaling is fully preserved in pericytes from patients with PAH.

### CNP attenuates the hyper-responsiveness of PAH pericytes to growth factors
In agreement with published studies[7,8], lung pericytes isolated from PAH patients had slightly enhanced basal proliferation rates, as analyzed by bromodeoxyuridine (BrdU) incorporation assays (Fig. 2a). Moreover, the proliferative effect of PDGF-BB (30 ng/ml, 24 h) was markedly greater in PAH than in control pericytes (Fig. 2a). Scratch assays demonstrated that this was associated with enhanced baseline and PDGF-BB-stimulated migration (Fig. 2b shows the progressive closing of the scratch-induced wound area during 24 h, as ratio of the starting wound area). Moreover, the expression of the PDGF receptor-β (PDGFR-β) was significantly upregulated in PAH compared to control pericytes (immunoblot in Supplemental Fig. 1), which most likely contributes to their augmented responses.

Next, we studied whether CNP attenuates the proliferative and pro-migratory effects of PDGF-BB. Because it was reported that growth factors desensitize the GC-B receptor in fibroblasts[16], firstly, we studied in control pericytes whether PDGF-BB influences the cGMP responses to CNP. The cells were pretreated with 30 ng/ml PDGF-BB for 15 min, followed by CNP stimulation for an additional 15 min. As shown in Fig. 2c, PDGF-BB slightly reduced the effects of 10 and 100 nM CNP on pericyte cGMP levels, but this interaction was statistically insignificant.

To dissect the effects of CNP on PDGF-BB–stimulated pericyte proliferation and migration, the cells were pretreated with 10 or 100 nM CNP for 30 min, followed by incubation with 30 ng/ml PDGF-BB for 24 h. CNP significantly attenuated PDGF-BB-induced proliferation of control (Fig. 2d, upper panel) and PAH pericytes (lower panel). As shown, this inhibitory effect was even stronger in PAH pericytes. Intriguingly, ANP (100 nM) tended to have the opposite effect, slightly enhancing the proliferative actions of PDGF-BB in control and especially in PAH pericytes (Fig. 2d). Corroborating these results of the BrdU assays, PDGF-BB markedly enhanced pericyte's expression of the cell cycle protein cyclin D1 and of the proliferation marker, proliferating cell nuclear antigen (PCNA) and CNP pretreatment attenuated this effect in control and PAH pericytes (Fig. 2e and f). Moreover, CNP significantly inhibited the promigratory effects of PDGF-BB in both groups of pericytes (Fig. 2g).

Whereas PDGF-BB has been implicated in the enhanced proliferation and migration of pericytes in lungs from patients with PAH, enhanced levels of TGF-β and augmented expression of TGF-β receptors contribute to their hypercontractile phenotype[7,8]. As shown in Fig. 3a, TGF-β pretreatment (10 ng/ml, 15 min) did not affect pericytes´ cGMP responses to CNP. Therefore, we studied whether CNP/cGMP signaling attenuates the induction of α-SMA by TGF-β. In control pericytes, TGF-β (10 ng/ml, 24 h) had very variable effects on the expression of α-SMA, which on average were not significant (Fig. 3b). In PAH pericytes, the expression of the TGF-β receptor II was slightly enhanced (Supplemental Fig. 1) and, concordantly, TGF-β consistently induced α-SMA expression (Fig. 3b). Pretreatment with 10 and 100 nM CNP for 30 min significantly attenuated this TGF-β effect in PAH pericytes (Fig. 3c).

Our results confirm that pericytes from PAH patients exhibit exacerbated responses to growth factors such as PDGF-BB and TGF-β. They add a novel and important piece of information, namely that CNP/cGMP signaling markedly attenuates the effects of these growth factors on pericyte proliferation, migration, and dedifferentiation.

### CNP attenuates PDGF-BB-induced pro-proliferative signaling pathways in human control and PAH lung pericytes
Binding of PDGF-BB to the PDGFR-β induces autophosphorylation of the receptor and activates a cascade of intracellular kinases, including AKT and extracellular signal-regulated kinases (ERK), ultimately increasing cell proliferation[17]. Therefore, next, we studied whether CNP prevents the activation of AKT and/or ERK by PDGF-BB. As before, cells were pretreated with CNP (10 and 100 nM, 30 min) followed by PDGF-BB (30 ng/ml, for an additional 30 min). Figure 4a shows that the CNP-induced phosphorylation of VASP at $Ser_{239}$ was preserved in the presence of PDGF-BB and

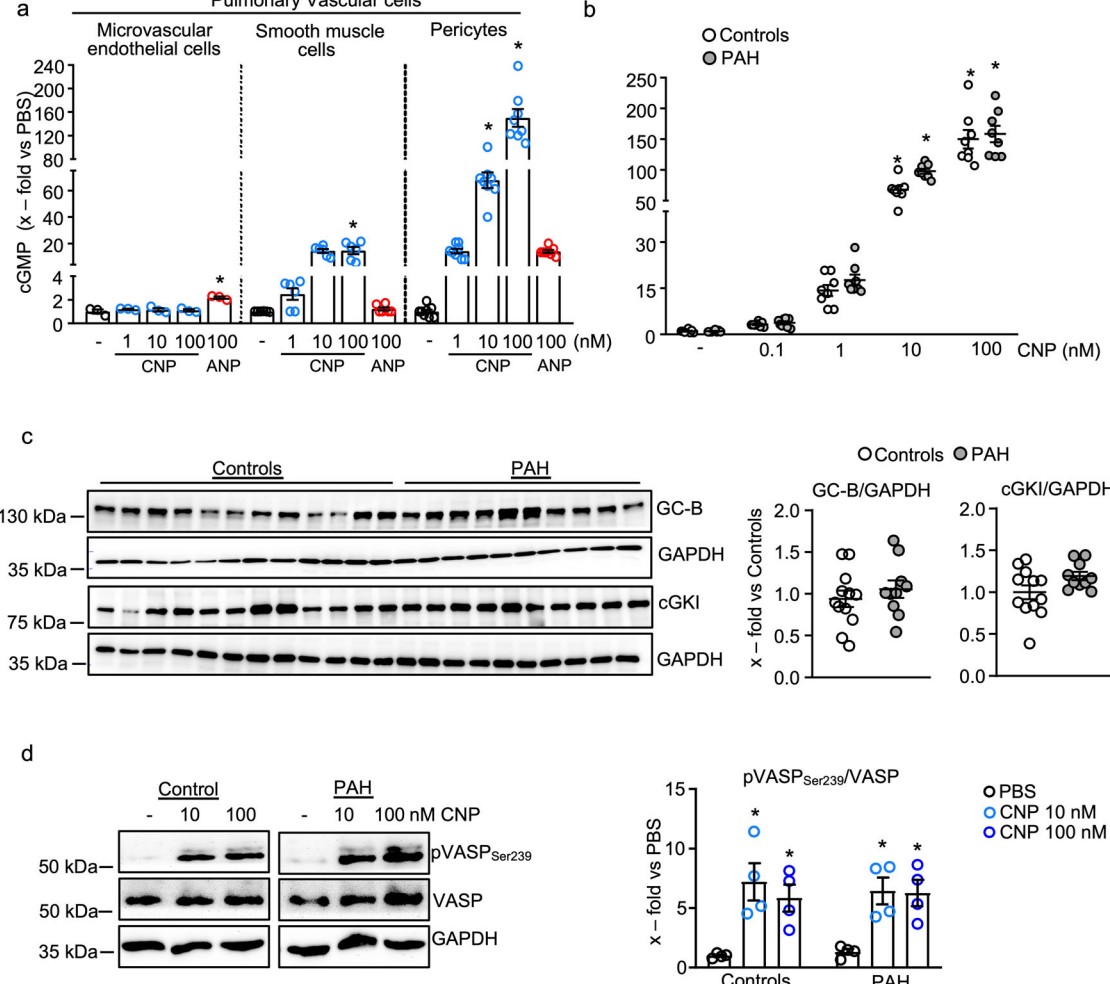

**Fig. 1 | Unaltered CNP/GC-B/cGMP signaling in pericytes from patients with pulmonary arterial hypertension (PAH). a** Effects of CNP and ANP on cGMP levels of human lung microvascular endothelial cells ($n = 3$ wells from one biological sample), vascular smooth muscle cells ($n = 6$ wells from three biological replicates), and pericytes ($n = 8$ wells from four biological replicates) (1-way ANOVA). **b** and **c** Lung pericytes isolated from control individuals and patients with PAH exhibit

similar cGMP responses to CNP and similar expression levels of the GC-B receptor and cGKI (**b**: $n = 8$ wells from four biological replicates (1-way ANOVA). **c** $n = 12$ from 6 controls and $n = 10$ from 5 PAH patients (unpaired 2-tailed Student's $t$-test). **d** CNP similarly stimulates the phosphorylation of VASP at $Ser_{239}$, the cGKI-specific site, in control and PAH pericytes ($n = 4$ biological replicates per group (2-way ANOVA)). *$P < 0.05$ vs. PBS (−).

unchanged in PAH pericytes, indicating unaltered activation of cGKI. As expected, PDGF-BB (30 ng/ml, 30 min) led to strong increases in the phosphorylation of AKT (at $Ser_{473}$) and ERK 1/2 (at $Thr_{202}/Tyr_{204}$) (Fig. 4b). Notably, CNP significantly prevented these effects in both control and PAH pericytes (Fig. 4b).

Among the many downstream targets of AKT and ERK signaling, the Forkhead box O (FoxO) transcription factors FoxO1 and FoxO3 have an important role in the pathogenesis of PAH[18,19]. Activation of AKT and/or ERK by PDGF-BB leads to FoxO phosphorylation at $Thr_{32}$ and thereby to its nuclear exclusion and downregulation[20–22]. This pathway contributes to the enhanced proliferation of PASMCs in PAH as well as of fibroblasts in lung fibrosis[18,21]. Since FoxO3 is the predominant FoxO isoform expressed in pericytes[23], we investigated whether CNP prevents PDGF-BB-mediated FoxO3 phosphorylation. PDGF-BB stimulation (30 min) led to a significant increase in the phosphorylation of FoxO3 at $Thr_{32}$ in both control and PAH pericytes, and CNP significantly reduced this effect (Fig. 5a). Remarkably, 100 nM CNP completely prevented the PDGF-BB-induced FoxO3 phosphorylation in PAH pericytes. In line with these results, immunocytochemistry showed that PDGF-BB (30 ng/ml during 6 h) significantly reduced nuclear FoxO3 localization in pericytes (Fig. 5b depicts quantification of nuclear fluorescence). Pretreatment with CNP (100 nM, 30 min) prevented this effect, increasing nuclear FoxO3 (Fig. 5b).

To determine whether FoxO3 participates in the antagonistic effects of PDGF-BB and CNP on pericyte proliferation, we tested the effect of siRNA-mediated FoxO3 knockdown. Immunoblotting revealed that FoxO3 siRNA (si-FoxO3) transfection for 48 h led to a significant, marked reduction of FoxO3 protein in comparison to non-transfected and control siRNA (si-Control) transfected pericytes (Fig. 5c). In the si-Control-transfected pericytes, PDGF-BB significantly increased proliferation and CNP prevented this effect (Fig. 5d). In contrast, in the FoxO3-deficient cells the baseline proliferation was slightly enhanced, and PDGF-BB only exerted a minor additional effect. We assume that the observed overall duplication of the proliferation rate is the maximal achievable effect in control pericytes (see also Fig. 2a). Notably, in such FoxO3 knock-down pericytes, the antiproliferative effect of CNP was fully abolished (Fig. 5d). Together these results suggest that CNP antagonizes proliferative PDGF-BB signaling in lung pericytes by inhibiting AKT and ERK activation and subsequent FoxO3 phosphorylation. This stabilizes nuclear FoxO3 expression and fosters the antiproliferative effects of this transcription factor.

**cGKI mediates the inhibitory effects of CNP on PDGF-BB-induced pericyte proliferation**
To elucidate whether the antiproliferative effect of the CNP/GC-B/cGMP pathway is mediated by activation of cGKI, we applied the specific cGKI

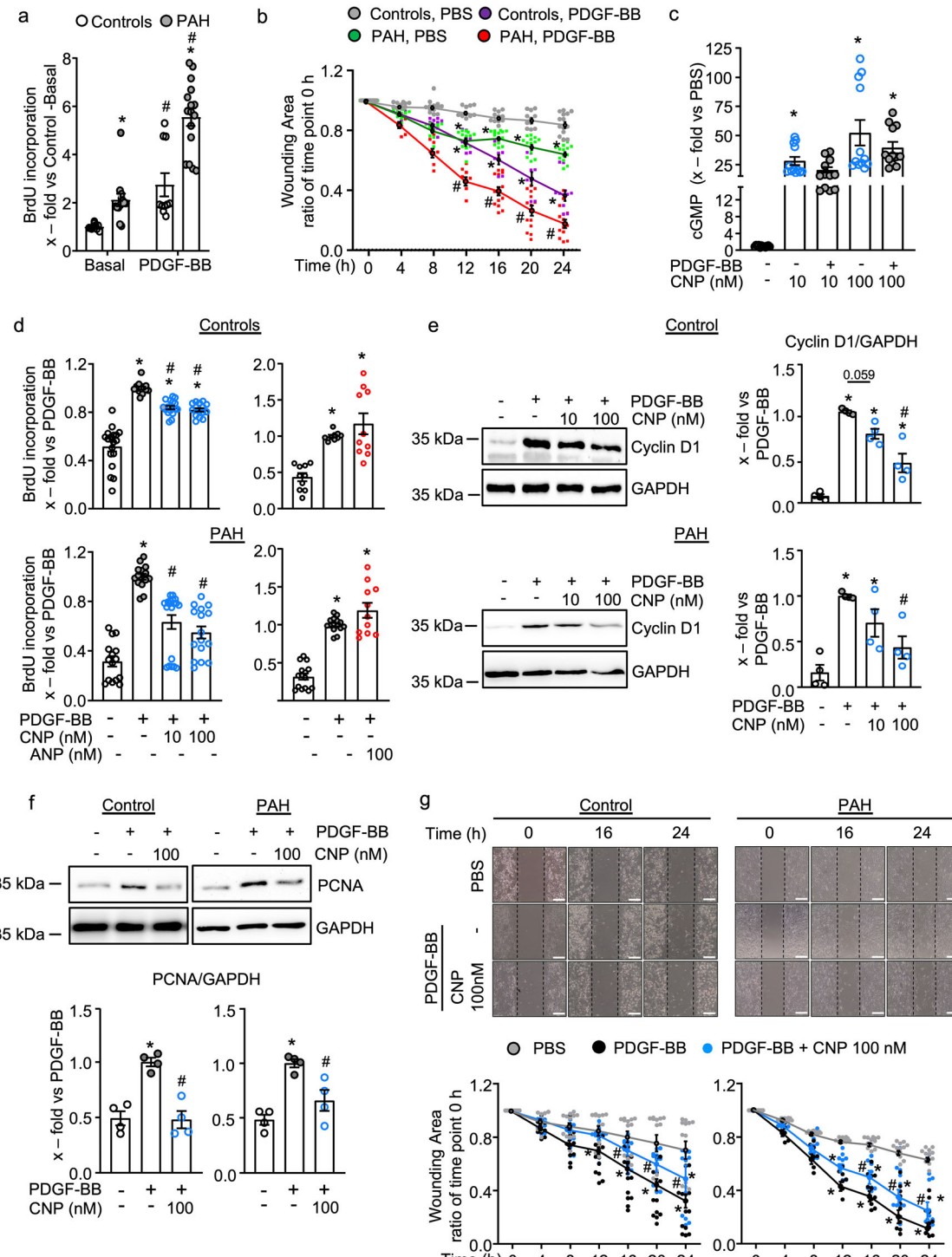

**Fig. 2 | CNP attenuates the stimulatory effects of PDGF-BB on pericyte proliferation and migration. a** and **b** In comparison to control pericytes, PAH pericytes had higher proliferation (**a**) and migration rates (**b**) at baseline and in response to PDGF-BB (30 ng/ml, 24 h) (**a** BrdU incorporation, *n* = 10 wells from three biological replicates for control and *n* = 16 wells from four biological replicates for PAH; non-parametric Kruskal–Wallis analyses. **b** Scratch assays with 8–14 wells from three biological replicates from each group; 2-way ANOVA). **c** PDGF-BB (30 ng/ml, 15 min pretreatment) did not significantly alter the cGMP responses of control pericytes to CNP (*n* = 12 wells from four biological replicates per condition; non-parametric Kruskal–Wallis analyses). **d–f** CNP (pretreatment for 30 min) attenuated PDGF-BB (30 ng/ml) induced proliferation (**d**), cyclin D1 (**e**) and PCNA

protein expression (**f**) of control and PAH pericytes (**d**: *n* = 10–18 wells from three biological replicates per group; non-parametric Kruskal–Wallis analyses; **e** and **f**: *n* = 4 biological replicates from control and PAH pericytes; 1-way ANOVA). **g** CNP (100 nM, 24 h) attenuated PDGF-BB-induced migration of control and PAH pericytes. Top panels: representative pictographs of control (left) and PAH pericytes (right) at 0, 16, and 24 h of the scratch assay (scale bar: 500 mm); Bottom panels, evaluation of the wounding areas in comparison to the initial wound (*n* = 8–13 wells from three biological replicates per group; 2-way ANOVA). For **a**: *$p < 0.05$ vs. Controls, #$p < 0.05$ vs. Basal. For **b**: *$p < 0.05$ vs. PBS-Control, #$p < 0.05$ vs. PDGF-BB-Control. For **c**: *$p < 0.05$ vs. PBS (−). For **d–g**: *$p < 0.05$ vs. PBS (−), #$p < 0.05$ vs. PDGF-BB.

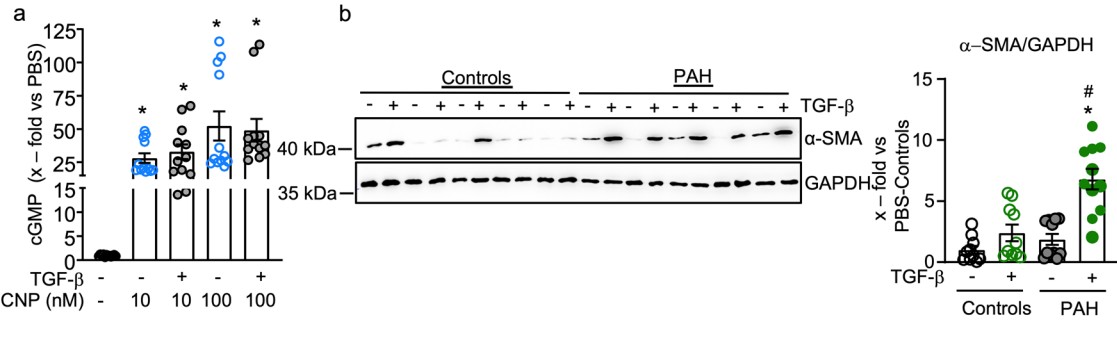

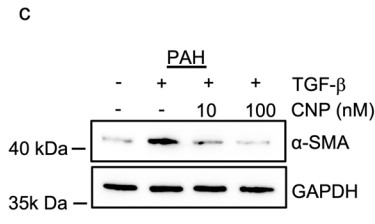

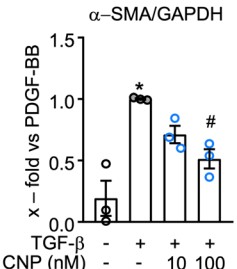

**Fig. 3 | CNP attenuates the stimulatory effects of TGF-β on α-SMA expression.**
**a** TGF-β (10 ng/ml, 15 min) did not significantly alter the cGMP responses of
control pericytes to CNP ($n = 12$ from four biological replicates; non-parametric
Kruskal–Wallis analyses). **b** Immunoblotting: TGF-β (10 ng/ml, 24 h) enhanced α-
smooth muscle actin (α-SMA) expression more in PAH pericytes than in controls
($n = 10$ from five biological replicates in each group; non-parametric

Kruskal–Wallis analyses). **c** Immunoblotting: Pretreatment with CNP (10 and
100 nM, 15 min) significantly attenuated TGF-β (10 ng/ml, 24 h)-induced α-SMA
expression in PAH pericytes ($n = 3$ biological replicates; 1-way ANOVA). For **a**:
*$p < 0.05$ vs. PBS (−). For **b**: *$p < 0.05$ vs. PBS (−), #$p < 0.05$ vs. TGF-β in controls. For
**c**: *$p < 0.05$ vs. PBS (−), #$p < 0.05$ vs. TGF-β.

inhibitor Rp-8-Br-PET-cGMPS. As shown in Supplemental Fig. 2, pre-
treatment of control pericytes with Rp-8-Br-PET-cGMPS (100 μM, 20 min)
fully prevented the effect of CNP on VASP phosphorylation at $Ser_{239}$ (the
cGKI-specific site), demonstrating efficient cGKI inhibition. Notably, such
inhibition of cGKI did not prevent the suppressing effect of CNP on PDGF-
BB-induced $ERK_{Thr202/Tyr204}$ phosphorylation (Supplemental Fig. 2).
However, it fully abolished the inhibitory effect of CNP on the PDGF-BB-
induced phosphorylation of AKT and FoxO3 (Fig. 6a, middle and right
panels). In line with these results, Rp-8-Br-PET-cGMPS (10 μM, 20 min)
did not alter the baseline or PDGF-BB-stimulated proliferation of pericytes
but fully prevented the counter-regulation of such PDGF-BB effects by CNP
(Fig. 6b and c depict BrdU incorporation assays and PCNA immunoblot-
ting, respectively).

To support these studies with the cGKI inhibitor, we also studied
whether 8-Bromo-cGMP, a specific cGKI activator, mimics the effects of
CNP. In the presence of PDGF-BB, 8-Bromo-cGMP (0.1–10 μM, 30 min)
increased $VASP_{Ser239}$ phosphorylation in concentration-dependent man-
ner, demonstrating cGKI activation (Supplemental Fig. 3a). Concordantly,
8-Bromo-cGMP significantly and concentration-dependently reduced the
stimulatory effects of PDGF-BB on AKT and FoxO3 phosphorylation as
well as on pericyte proliferation (Supplemental Fig. 3b and c). These results,
taken together, clearly demonstrate that the inhibitory effects of CNP/GC-
B/cGMP signaling on the proliferative PDGF-BB/AKT/FoxO3 pathway are
mediated by cGKI. But how can the activation of a kinase prevent the
phosphorylation of AKT?

Phosphatase and tensin homolog (PTEN) is a phosphatase that inhi-
bits the PI3K/AKT pathway. Phosphorylation of PTEN at $Ser_{380}/Thr_{382/383}$
by Rho-associated kinase 1 (ROCK1) or Glycogen synthase kinase-3 beta
(GSK-3β) reduces its phosphatase activity[22,24]. On the other hand, cGKI
phosphorylates RhoA at $Ser_{188}$ (essential for activation of ROCK1) and
GSK-3β at $Ser_9$, resulting in their inactivation[25,26]. Therefrom, we hypo-
thesized that CNP, via a cGKI-mediated inhibition of RhoA and/or GSK-3β,
leads to PTEN activation and thereby attenuates the effects of PDGF-BB on
the PI3K-AKT axis (a scheme illustrating this pathway is depicted in Fig. 7a).
To follow up this hypothesis we studied the phosphorylation of PTEN at

$Ser_{380}/Thr_{382/383}$. As shown in Fig. 7b, incubation of control pericytes with
PDGF-BB (30 ng/ml, 30 min) led to an increase in PTEN phosphorylation
at $Ser_{380}/Thr_{382/383}$, which was prevented by CNP (100 nM, 30 min pre-
treatment). Inhibition of cGKI with Rp-8-Br-PET-cGMPS evoked a mild
increase of PTEN phosphorylation at baseline, which was further increased
by PDGF-BB (Fig. 7b). As also shown, the inhibitory effect of CNP on
PDGF-BB-driven PTEN phosphorylation was fully prevented. Conversely,
activation of cGKI with 8-Bromo-cGMP (0.1–10 μM, 20 min pretreatment)
prevented PDGF-BB-induced PTEN phosphorylation, thereby mimicking
the effects of CNP (Fig. 7c). Our observations indicate that the CNP/cGMP/
cGKI pathway indirectly leads to the activation of the phosphatase PTEN,
thereby attenuating the PDGF-BB induced AKT phosphorylation and
AKT-mediated inactivation of FoxO3 (scheme in Fig. 7a).

## Reduced pulmonary CNP mRNA expression in experimental and clinical pulmonary hypertension

Taken together, our results demonstrate that *exogenous*, synthetic CNP,
via GC-B/cGMP signaling, counteracts the growth factor-induced pro-
liferation, migration, and transdifferentiation of pericytes from patients
with PAH. This raises the question of whether the *endogenous* endo-
thelial hormone exerts local protective effects in pulmonary micro-
circulation and whether such effects are preserved in PAH. To approach
this question, firstly, we studied lung CNP and GC-B expression levels in
two experimental models of PH: Monocrotaline (MCT)-induced PH in
rats[27] and milder, chronic hypoxia (HOX)-induced PH in mice[28].
Quantitative real-time RT-PCR (qRT PCR) revealed that the CNP
expression levels were significantly reduced in lung samples from rats
and mice with PH in comparison to respective controls (Fig. 8a and b, left
panels). As also shown, GC-B expression was unchanged in MCT rats,
while it was significantly reduced in HOX-exposed mice in comparison
to control lungs (Fig. 8a and b, right panels). To follow up on the possible
clinical relevance, we also studied lung samples from PAH patients. In
line with the experimental models, CNP levels were significantly reduced
in human PAH as compared to control lungs, whereas GC-B levels were
unaltered (Fig. 8c).

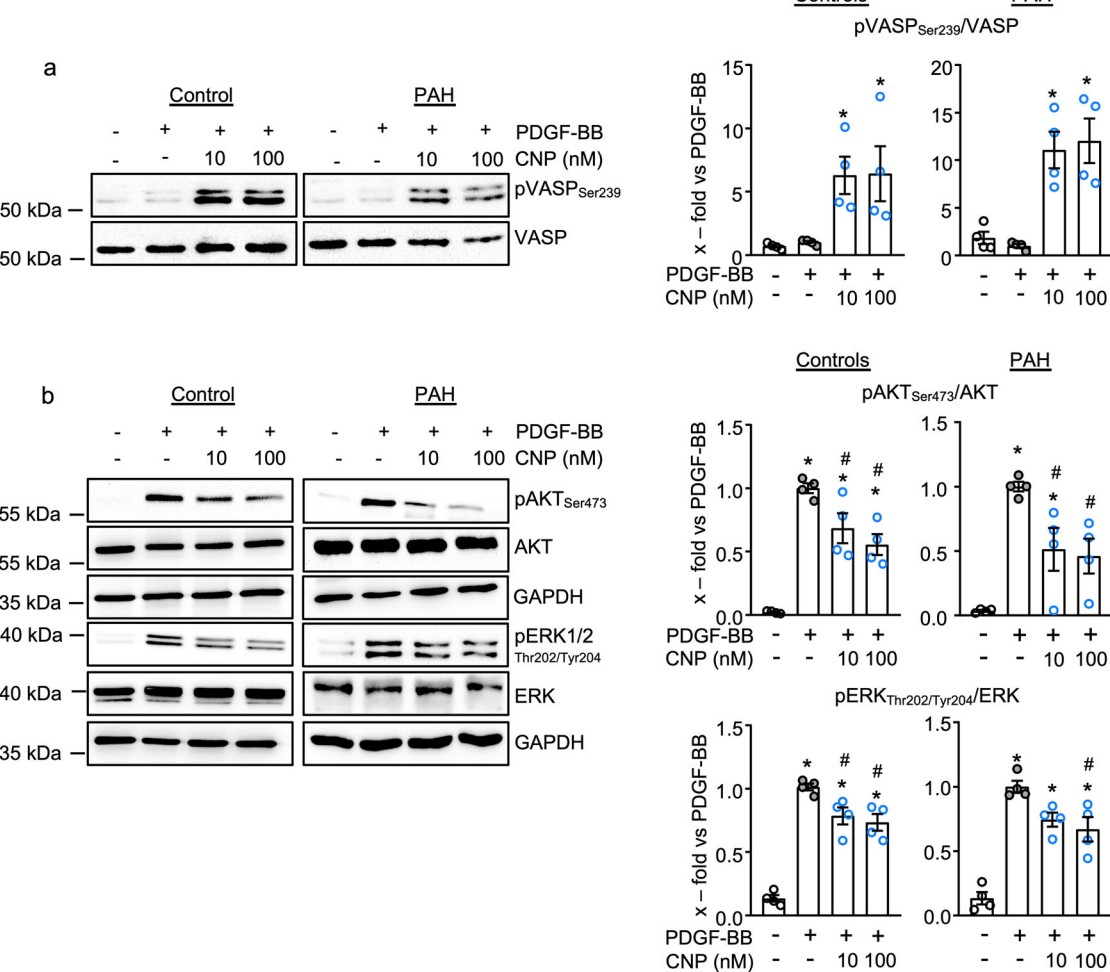

**Fig. 4 | CNP attenuates PDGF-BB-induced pro-proliferative signaling pathways in human control and PAH lung pericytes. a** CNP (10 and 100 nM, 15 min) increased VASP (Ser$_{239}$) phosphorylation in PDGF-BB-pretreated control ad PAH pericytes. **b** Pretreatment with CNP prevented the PDGF-BB (30 ng/ml, 30 min)- induced phosphorylation of AKT (Ser$_{473}$) and ERK1/2 (Thr$_{202}$/Tyr$_{204}$) in control and PAH pericytes ($n = 4$ from control and PAH pericytes; 1-way ANOVA). $^*p < 0.05$ vs. PBS (−), $^#p < 0.05$ vs. PDGF-BB.

## Discussion

Pericyte dysfunction, with excessive migration, ectopic proliferation, and differentiation into a smooth muscle-like cell, plays an important role in the progression of PAH[3,6,9]. Our studies reveal that *exogenous* CNP counter-regulates such growth factor-induced pericyte alterations by preserving the nuclear activity of the transcription factor FoxO3. Reduced expression of *endogenous*, paracrine-acting CNP might, therefore, contribute to the progression of vascular remodeling in patients with PAH.

CNP exerts its auto/paracrine effects by binding to its cGMP-producing receptor, GC-B[29]. In the lung, CNP is produced by ECs[30], and GC-B is expressed in different types of cells[31]. CNP infusion or over-expression of GC-B prevented lung vascular remodeling in preclinical models of PH[14,15]. However, the targeted pulmonary vascular cells, signaling mechanisms, and potential clinical implications were unclear. Our present studies reveal that *exogenous* synthetic CNP markedly activates GC-B/cGMP signaling in human lung pericytes and much less in PASMCs and ECs. Interestingly, the effect of ANP on lung pericyte cGMP levels was much weaker than the effect of CNP, while ECs showed greater responses to ANP. Hence, CNP signaling prevails in pericytes and ANP signaling in ECs, which emphasizes the distinct and complementary functions of these hormones. In fact, our previous studies had already shown endothelial-dependent protective effects of ANP in experimental PH[28].

While the expression and function of protective pathways often decline in diseased cells, GC-B expression and signaling were fully preserved in pericytes from PAH patients. Moreover, growth factors such as PDGF-BB and TGF-β, known to be upregulated in PAH and to desensitize the GC-B receptor in fibroblasts[16,32], did not alter the responsiveness of GC-B to CNP nor downstream cGMP/cGKI signaling in lung pericytes.

Augmented PDGF-BB and TGF-β release and signaling contribute to vascular remodeling and inflammation in PAH[3,33–35]. Pericytes isolated from patients with PAH exhibited higher PDGFR-ß expression and, concomitantly, increased proliferation and migration under PDGF-BB stimulation ([8] and present manuscript). As shown here, CNP diminishes PDGF-BB-induced proliferation and migration of lung pericytes, with even stronger antiproliferative effects in PAH pericytes. Inhibition of cGKI prevented the effects of CNP on PDGF-BB-induced proliferation, whereas 8-Bromo-cGMP, an activator of cGKI, mimicked the effects of CNP. Together these data demonstrate that the antiproliferative effects of CNP are mediated by cGKI and that CNP/GC-B/cGMP/cGKI signaling is unaltered in PAH pericytes.

PAH pericytes also exhibit higher expression of TGF-β receptor II and in vitro studies have shown that TGF-β, by enhancing the expression of α-SMA, promotes pericyte differentiation into smooth muscle-like cells ([6] and present studies). As shown here, CNP markedly attenuated the TGF-β-induced α-SMA expression in pericytes from PAH patients. Based on studies in cardiac fibroblasts, this is possibly mediated by the inhibitory effects of cGMP/cGKI signaling on TGF-β/Smad signaling[36,37]. Phosphorylation of Smad3 at Ser$_{309}$/Thr$_{368}$ by cGKI prevented its heterodimerization with

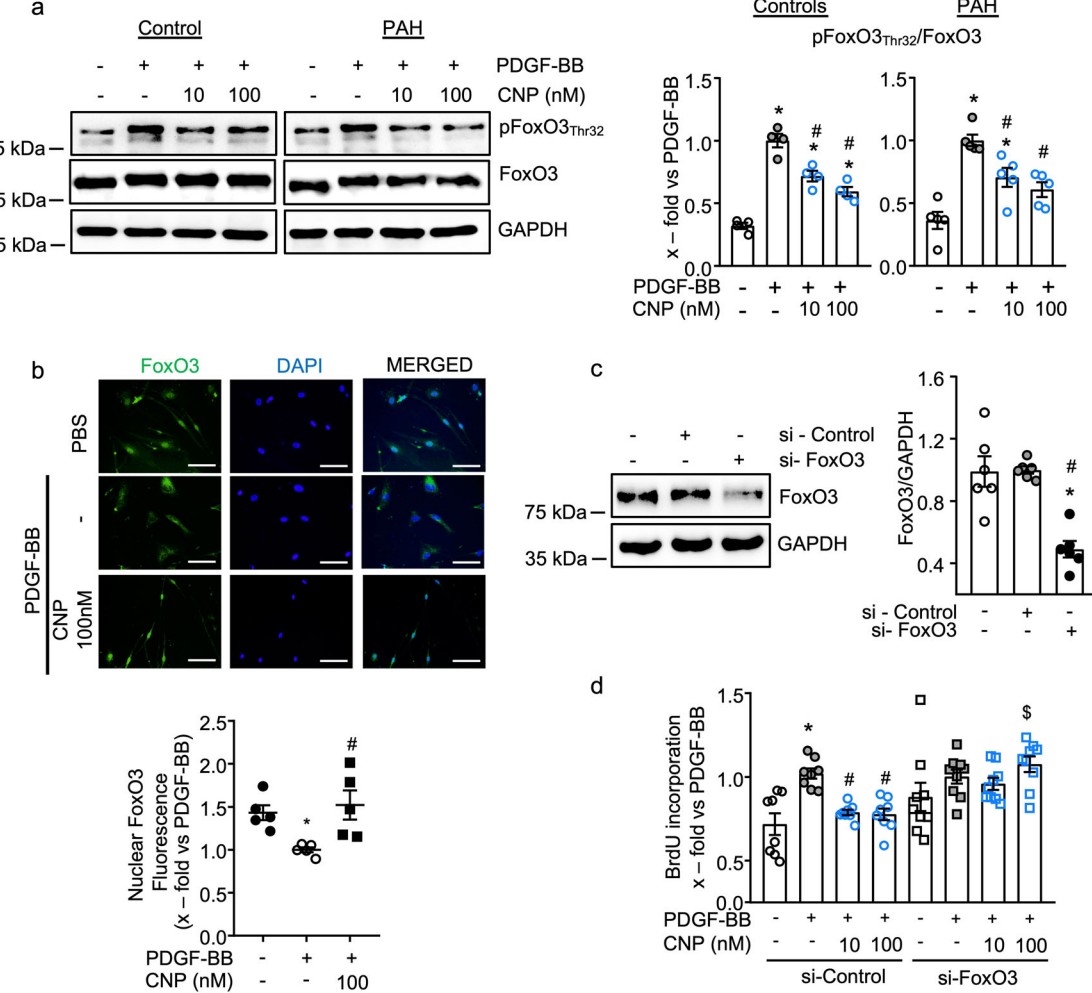

**Fig. 5 | FoxO3 mediates the counter-regulation of PDGF-BB-induced pericyte proliferation by CNP. a** CNP (10 and 100 nM, 15 min) prevented the PDGF-BB (30 ng/ml, 30 min)-induced phosphorylation of FoxO3 (Thr$_{32}$) in control and PAH pericytes ($n$ = 4 control pericyte samples, $n$ = 5 from four PAH pericytes; 1-way ANOVA). **b** CNP (100 nM) pretreatment prevented the PDGF-BB-induced FoxO3 nuclear exclusion as assessed by immunocytochemical staining of FoxO3, followed by nuclear fluorescence intensity measurement by Image J ($n$ = 5 from three biological replicates; each value is the mean of three images; 1-way ANOVA). Scale bar: 100 mm. **c** and **d** Transfection of control pericytes with siFoxO3 reduced FoxO3 protein expression (**c**: $n$ = 6 independent experiments from three biological replicates; 1-way ANOVA), and this prevented the inhibitory effects of CNP on PDGF-BB induced pericyte proliferation (**d**: $n$ = 9 wells from three biological replicates; 2-way ANOVA). For **a** and **b**: *$p$ < 0.05 vs. PBS (−), #$p$ < 0.05 vs. PDGF-BB. For **c**: *$p$ < 0.05 vs. untransfected control (−), #$p$ < 0.05 vs. si-Control. For **d**: *$p$ < 0.05 vs. PBS (−), #$p$ < 0.05 vs. PDGF-BB, $$p$ < 0.05 vs. corresponding vehicle-treated group (−).

Smad4 and interrupted their nuclear translocation, resulting in repression of transcriptional activation by TGF-β[36,37].

The proliferative effects of PDGF-BB are mainly mediated by activation of the intracellular PI3K-AKT and MAPK/ERK pathways[38]. Subsequent phosphorylation and inactivation of the transcription factor Forkhead Box O1/3 (FoxO1/3) has a critical role in the increased proliferation of PASMCs and pulmonary fibroblasts, driving vascular wall thickening in patients with PAH[18,19]. Inhibition of these signaling pathways with tyrosine kinase inhibitors such as imatinib is efficiently used in the therapy of patients with PAH, although with severe unwarranted effects[39]. Interestingly, the present study reveals that CNP diminishes PDGF-BB-induced AKT phosphorylation (at Ser$_{473}$) and concomitant FoxO3 phosphorylation (at Thr$_{32}$) and, thereby, FoxO3 nuclear exclusion, indicating that FoxO3 activation mediates the anti-proliferative CNP effects. Indeed, siRNA-mediated FoxO3 knockdown fully prevented this action of CNP.

Previous studies had already implicated CNP in AKT regulation, albeit in a cell type-specific manner. In HUVECs, CNP increased AKT phosphorylation at Ser$_{473}$ and thereby attenuated lipopolysaccharide-induced endothelial dysfunction[40]. A similar AKT-stimulating CNP effect was observed in lung microvascular ECs and was attributed to activation of a

second specific CNP receptor, the natriuretic peptide receptor C (NPR-C)[11]. Together with our present results, this suggests that CNP/NPR-C signaling activates AKT, while the CNP/GC-B–cGMP axis inhibits AKT phosphorylation. Hence, different types of cells may exhibit distinct responses to CNP depending on their relative expression levels of NPR-C and GC-B. Corroborating this notion, in our studies of lung pericytes, which have high GC-B levels, CNP prevented PDGF-BB stimulated AKT and FoxO3 phosphorylation through cGMP/cGKI activation. Similarly, in intestinal epithelial cells, the activation of the guanylyl cyclase-C (GC-C) receptor resulted in high intracellular cGMP levels and cGMP-mediated inhibition of AKT signaling[41]. This effect was attributed to the activation of the phosphatase PTEN, which dephosphorylates phosphatidylinositol-3,4,5-triphosphate (PIP3) and thereby inhibits the PI3K–AKT pathway[42]. In line with these published observations, in lung pericytes, PDGF-BB elicited an inhibitory phosphorylation of PTEN at Ser$_{380}$/ Thr$_{382/383}$[24], and this effect was prevented by CNP. Our data indicate that CNP, via GC-B/cGMP/cGKI signaling, prevents PDGF-dependent AKT phosphorylation by preserving or enhancing the phosphatase activity of PTEN.

How does cGMP/cGKI signaling activate PTEN? Based on published studies, we postulated an indirect pathway involving cGKI-dependent

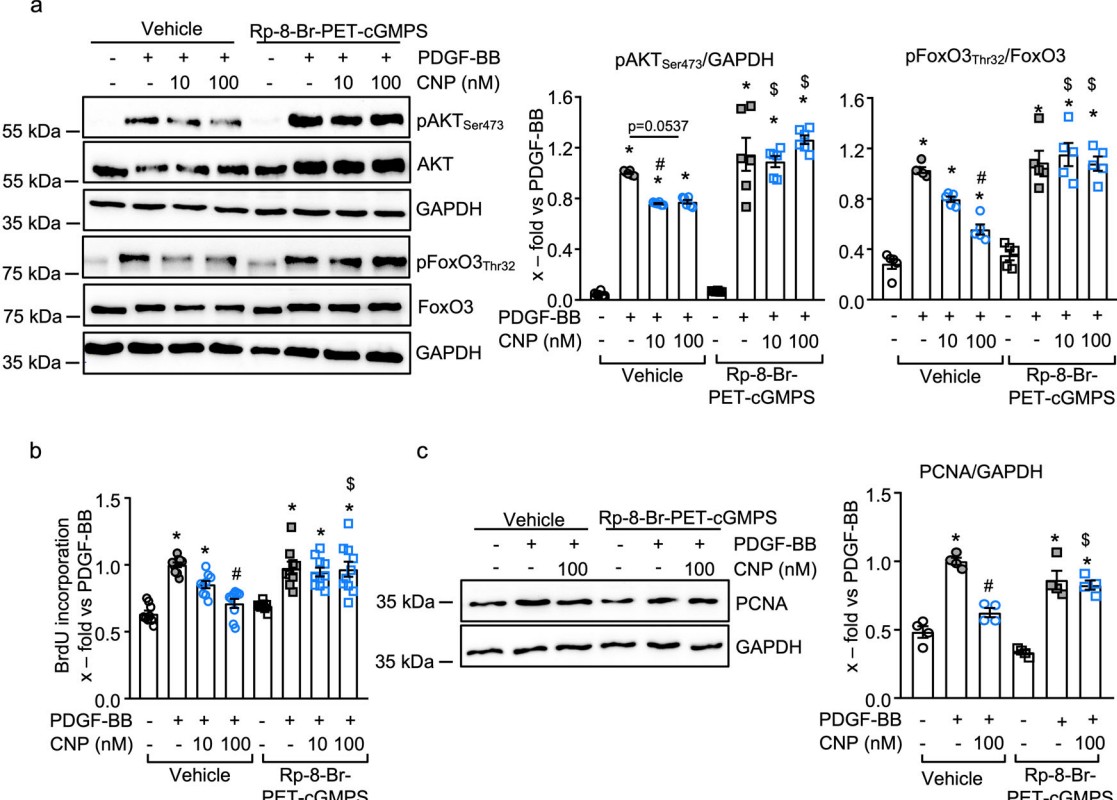

**Fig. 6 | Inhibition of cGKI prevents the effects of CNP on PDGF-BB-induced proliferative signaling. a** The cGKI inhibitor Rp-8-Br-PET-cGMPS (100 μM) prevented the effect of CNP on PDGF-BB (30 ng/ml, 30 min)-induced phosphorylation of AKT (Ser$_{473}$) and FoxO3 (Thr$_{32}$) ($n = 5$–6 from four biological replicates; 2-way ANOVA). **b** and **c** Rp-8-Br-PET-cGMPS (10 μM) attenuated the inhibitory effects of CNP on PDGF-BB-induced proliferation of control pericytes as analysed by **b** BrdU incorporation assay ($n = 9$–10 wells from three biological replicates; non-parametric Kruskal–Wallis analyses) and **c** immunoblotting for PCNA ($n = 4$ biological replicates; 2-way ANOVA). *$p < 0.05$ vs PBS (−), #$p < 0.05$ vs. PDGF-BB,

$p < 0.05$ vs. corresponding vehicle-treated group. Please note that in the samples derived from cells treated with Rp-8-Br-PET-cGMPS the immunoreactive signal obtained for total AKT was markedly increased (second lane in the original westerns depicted in (**a**)). Presently, we do not have an explanation for this reproducible observation. Due to the brief incubation time (<1 h), we believe that this enhanced immunoreactivity is not derived from increased AKT protein expression but related to changes in AKT conformation and enhanced binding of the anti-AKT antibody to its epitope. Due to these changes, the signal of pAKT$_{Ser473}$ was normalized to GAPDH and not to total AKT (middle panel in Fig. 6a).

inhibition of ROCK1 and/or GSK3β, kinases that are known to elicit inhibitory phosphorylations of PTEN (an illustrating scheme is provided in Fig. 7a)[23,27]. It was shown that cGKI phosphorylates RhoA at Ser$_{188}$, thereby preventing ROCK1 activation[25]. In addition, cGKI also inhibits GSK3β[26]. Supporting the role of this mechanistic link in human lung pericytes, cGKI activation by the membrane-permeant analog 8-Bromo-cGMP attenuated PDGF-BB-induced PTEN phosphorylation. Conversely, inhibition of cGKI prevented the effects of CNP on PTEN. Therefrom, we conclude that CNP, via the GC-B/cGMP/cGKI pathway, indirectly preserves the activity of PTEN and thereby attenuates PDGF-BB/AKT signaling (see Fig. 7a).

As mentioned in the introduction, CNP infusion attenuated vascular remodeling in experimental PH[14,15]. Based on the here presented results, this protective effect of "*exogenous*" CNP is mediated in an important part by the stabilization of microvascular pericytes. Hence, it is possible that *endogenous* CNP mediates a physiological local communication from ECs to adjacent pericytes, which might be altered in PAH. The plasma levels of N-terminal proCNP were slightly increased in PH patients but exhibited a stronger association with renal function than with pulmonary circulation variables[43]. Moreover, considering the local and paracrine nature of CNP actions, circulating CNP levels possibly do not reflect its concentrations within the pulmonary endothelial-pericyte interphase. Therefore, we analyzed the CNP mRNA expression in lung samples from PAH patients and from animals with experimental, MCT- or HOX-induced PH. Notably, CNP mRNA levels were strongly reduced in the lungs of PAH patients and PH animals, which is in line with previous studies in chronic hypoxic rats[30]. In contrast, as shown here, the

GC-B expression levels were unaltered in PAH patients along with MCT rats, which is consistent with the preserved GC-B expression and signaling in cultured human PAH pericytes. Interestingly, GC-B was significantly downregulated in HOX mice lungs hinting towards a direct effect of HOX on CNP/GC-B signaling.

The observation of diminished CNP but preserved GC-B in hypertensive lungs (PAH patients and MCT rats) together may have pathophysiological and therapeutic implications. The inhibition of a CNP-mediated paracrine endothelial–pericyte communication will augment the responses of pericytes to growth factors (present studies) and vasoconstrictors[13], thereby contributing to the progression of the vascular alterations involved in PAH. Substitution of the endogenous hormone by exogenous peptide may have protective vascular effects, as already suggested by preclinical studies[14,15]. In these studies, high-dosed CNP was continuously infused through osmotic minipumps. Such treatment schedules, of course, are not feasible in patients with chronic PAH. A first step towards the clinical application of CNP was made with the approval of Vosoritide (VOXZOGO^R), a stabilized analog, for the treatment of children with achondroplasia[44]. The short $t_{1/2}$ of ≈30 min requiring daily injections, the high $C_{max}$ of ≈5 nM (with risk of arterial hypotension), and the high daily peak-to-through variations of plasma concentrations stimulated the development of the longer-acting CNP analog [Gln$^{6, 14}$]-CNP-38, equipotent to CNP, which after s.c. injections in animals had a half-life of nearly 1 month[44]. Simulations in humans indicated that this conjugate would be feasible, practical, and efficient for weekly, 2-weekly, and even monthly s.c. dosing[44], a treatment schedule that would be realistic in patients with PAH.

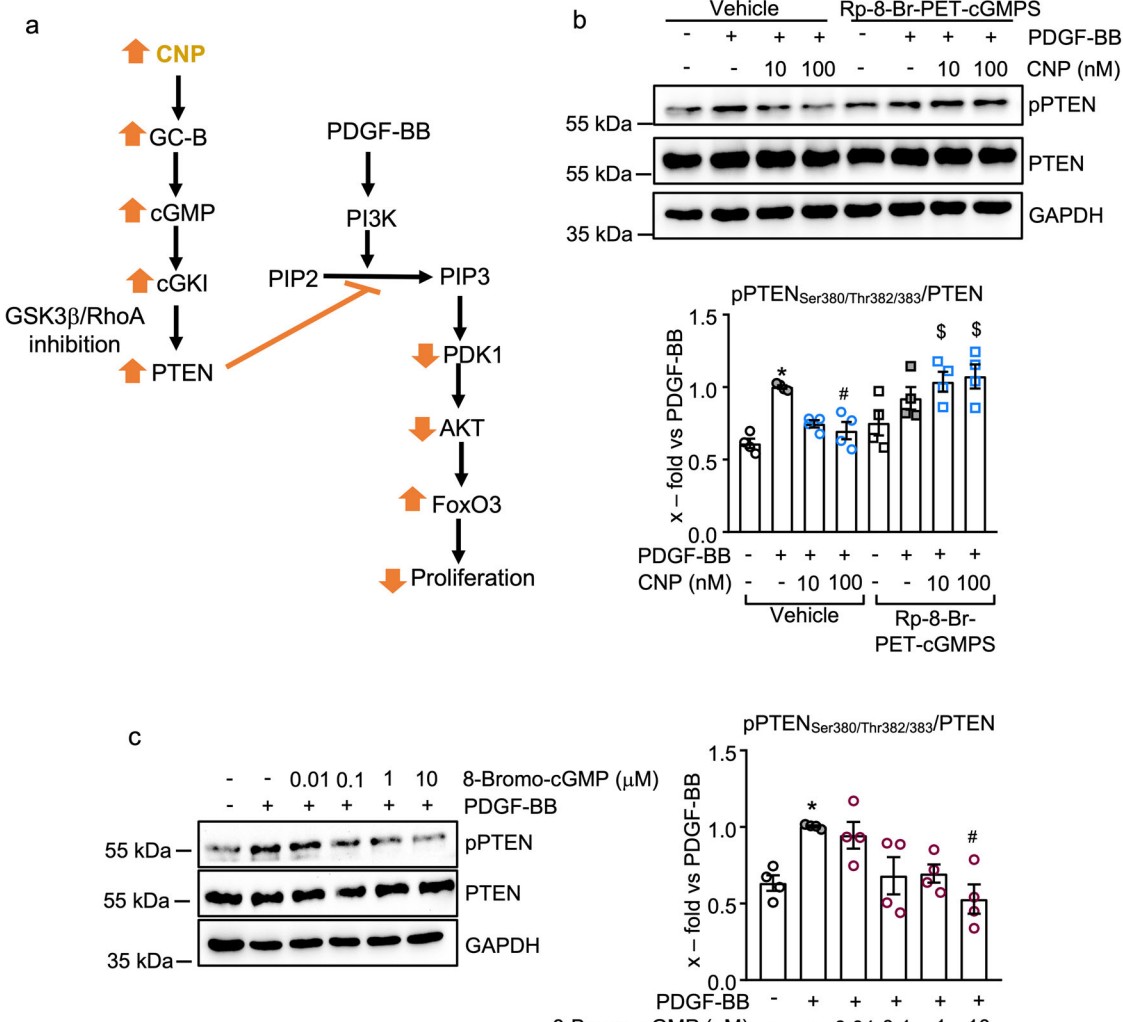

**Fig. 7 | CNP/cGKI signaling activates the phosphatase PTEN, thereby inhibiting PDGF-BB-mediated AKT phosphorylation. Fig. 7. a** Scheme of the postulated and investigated signaling pathway. PDGF-BB, via its PDGFR- β, triggers the activation of the PI3K, which phosphorylates PIP2 to PIP3. PIP3 then activates PDK1-AKT signaling. Subsequently, AKT phosphorylates and inactivates FoxO3, which enhances lung pericyte proliferation. CNP, via GC-B/cGMP signaling, activates cGKI. It has been shown that cGKI elicits inactivating phosphorylations of RhoA at Ser[188] and of GSK3b at Ser[9], thereby preventing their inhibitory phosphorylations of PTEN[27,28]. Activated PTEN dephosphorylates PIP3 and prevents AKT activation, resulting in an increase of nuclear FoxO3 and a concomitant reduction in pericyte proliferation. **b** PDGF-BB stimulated the phosphorylation of PTEN at Ser[380]/Thr[382/383] and CNP prevented this effect in the absence (vehicle) but not in the presence of

the cGKI inhibitor Rp-8-Br-PET-cGMPS (100 μM). **c** The cGKI activator, 8-Bromo-cGMP (0.01–10 μM), prevented PDGF-BB (30 ng/ml)-induced phosphorylation of PTEN in control pericytes (**b** and **c**: $n = 4$ biological replicates; **b**: 2-way ANOVA; **c**: 1-way ANOVA). $*p < 0.05$ vs. PBS (–), $^{\#}p < 0.05$ vs. PDGF-BB, $^{\$}p < 0.05$ vs. corresponding vehicle-treated group. PDGF-BB platelet-derived growth factor-BB, PDGFR-β platelet-derived growth factor beta, PI3K phosphoinositide 3-kinase, PIP2 phosphatidylinositol 4,5-bisphosphate, PIP3 phosphatidylinositol 3,4,5-tri-sphosphate, PDK1 phosphoinositide-dependent kinase 1, AKT protein kinase B, FoxO3 forkhead box O3, CNP C-type natriuretic peptide, GC-B guanylyl cyclase-B, cGMP cyclic guanosine monophosphate, cGKI cGMP-dependent kinase I, RhoA Ras homolog family member A, PTEN phosphatase and tensin homolog.

Our study allows the following main conclusions: (i) among human lung vascular cells, pericytes exhibit the strongest cGMP response to CNP, and this response is preserved or even slightly augmented in PAH pericytes; (ii) CNP effectively inhibits growth factor-induced proliferation migration, and dedifferentiation of control and PAH pericytes; (iii) ANP lacks these actions, highlighting the distinct effects of both "natriuretic peptides" on intracellular signaling pathways and cell functions; (iv) CNP exerts its antiproliferative effect through inhibiting the activation of AKT by PDGF-BB, which results in nuclear FoxO3 stabilization; cGKI-dependent PTEN activation participates in this pathway; and lastly, CNP expression is reduced in PAH lungs suggesting downregulation of protective endogenous CNP signaling.

Our future studies will be directed to further dissecting the local (patho) physiological roles of endogenous endothelial CNP in the lung as well as the factors and mechanisms provoking its downregulation in PH.

## Methods

### Clinical data of the PAH patients
The clinical information of the patients with PAH, from whom the lung tissues were obtained for pericyte isolations, is available in a previous publication of Prof. Perez[45].

### Isolation of lung pericytes
All ethical regulations relevant to human research participants were followed. Written informed consent was obtained from each individual patient or the patient's next of kin.

Human lung pericytes from control ($n = 3$) and PAH lungs ($n = 5$) were provided by Prof. Perez. The study protocol for tissue donation was approved by the ethics committee (Panel on Medical Human Subjects) of Stanford University (Stanford, USA) in accordance with national law and with the Good Clinical Practice/International Conference on

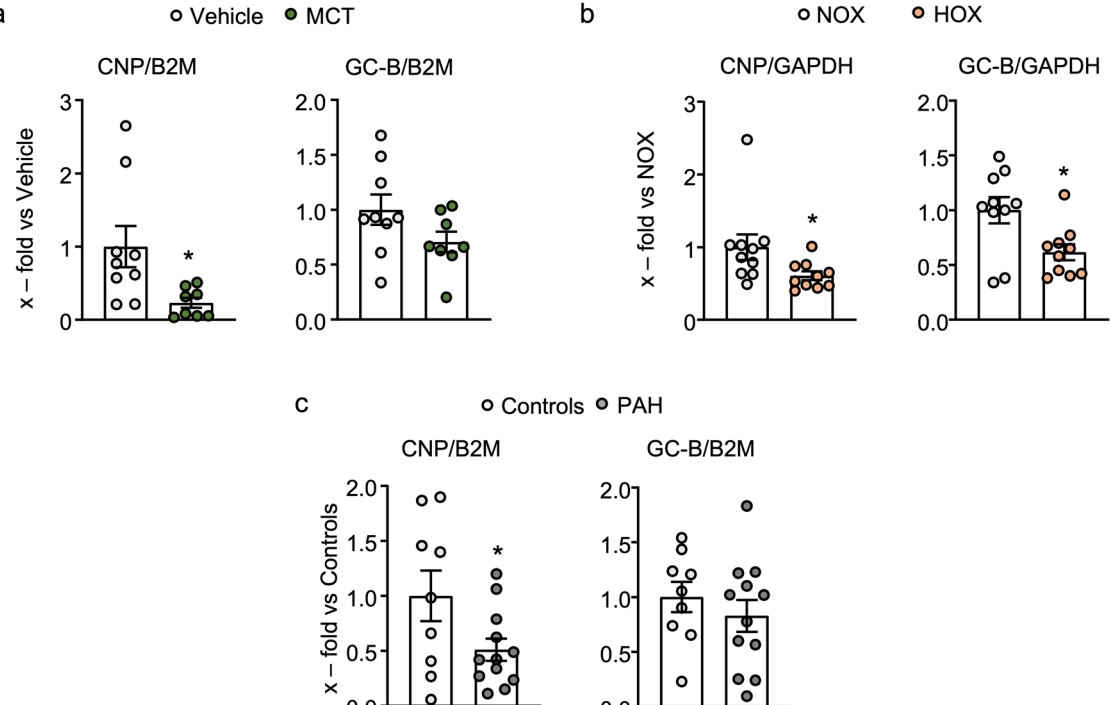

**Fig. 8 | Pulmonary CNP mRNA expression is reduced in clinical PAH and experimental PH. a** and **b** Lung CNP and GC-B mRNA expression in **a** monocrotaline (MCT) vs. vehicle-treated rats, **b** normoxia (NOX) vs. Hypoxia (HOX: 21 days) exposed mice and **c** PAH patients vs. controls. Values are the ratios of CNP or GC-B mRNA level relative to GAPDH (**b**) or b$_2$ microglobulin (**a** and **c**), determined by qRT-PCR and expressed as x-fold versus vehicle-treated rats (**a**), NOX mice (**b**), or human control lungs (**c**) (**a**: $n = 9$ samples from the vehicle and $n = 8$ from MCT-treated rats; **b**: $n = 10$ samples each from NOX vs. HOX exposed mice; **a** and **b**: Mann–Whitney test for CNP and unpaired 2-tailed Student's $t$-test for GC-B; **c**: $n = 9$ samples from controls and $n = 12$ samples from PAH patients, unpaired 2-tailed Student's $t$-test). *$p < 0.05$.

Harmonization guidelines. The study protocol is called 'Donating Unneeded Lung Tissue, Removed During Surgery, and Blood, for Medical Research", and the IRB # is 54182. The details of the isolation protocol and characterization are provided in the previous publication[45].

Lung pericytes from human control donors with lung cancer were isolated from tumor-free parts of anatomical resection specimens from the Department of Thoracic Surgery, University Hospital Wurzburg. The study protocol for tissue donation was approved by the ethics committee/Ethikkommission der Universität Würzburg in accordance with national law and with Good Clinical Practice/International Conference on Harmonization guidelines (Ref no. 20220831 02).

For isolation, lung tissue was cut and digested with an enzymatic solution consisting of Dispase (494207001, Sigma), Liberase (5401119001, Sigma), and DNase (A3778, Applichem). The resulting single-cell suspension was sequentially passed through 70 and 40 μm cell strainers, followed by red blood cell lysis. The cell suspension was incubated sequentially for 15 min with biotin labeled anti-PDGFRα (130-115-236, Miltenyi Biotec), anti-CD144 (130-100-754, Miltenyi Biotec), and anti-NG2 (130-116-374, Miltenyi Biotec) antibodies followed by 15 min incubation with anti-biotin magnetic microbeads (130-105-637, Miltenyi Biotec). Antibody-labeled cells were separated from the cell suspension by magnetic assisted sorting employing MS columns (130-042-201, Miltenyi Biotec) according to the manufacturer's protocol. PDGFR- α⁻ CD144⁻ NG2⁺ cell population was plated on cell culture dishes as pericytes.

Cells were characterized by positive staining for NG2 and PDGFR-β (pericyte markers) and negative staining for CD31 (endothelial marker) and PDGFR-α (fibroblast marker).

### Primary cell culture
Lung pericytes were maintained in Pericyte medium (sc-1201, Provitro) supplemented with growth factors and 2% fetal bovine serum. All experiments were carried out between passages 7–11 from a minimum of three subjects.

Each experiment was repeated a minimum of three times, and each time, cells from different human subjects were used to attest for biological heterogeneity. For functional experiments (BrdU incorporation Assay, Wound healing assay), minimum three technical replicas were used within one experiment.

Human pulmonary artery smooth muscle cells (PASMCs) were purchased from Lonza (CC-2581) and cultured in smooth muscle cell growth medium (CC-3182, Lonza) according to the protocol previously described[46]. All experiments were carried out at passage 6 from a minimum of three subjects.

Human microvascular lung endothelial cells (MLVECs) were purchased from Promocell (c-12281). The cells were maintained in microvascular endothelial medium MV (c-22120, Promocell) according to the manufacturer's instructions. RIA was carried out in passage 5.

### Treatment of cultured pericytes
Before experiments, cultured cells were incubated for 24 h in basal media with 0.1% FCS (serum-reduced medium). Pericytes were pre-treated with CNP (4095840: Bachem) or 8-Bromo-cGMP (B 004-50: Biolog) for 30 min, followed by stimulation with 30 ng/ml PDGF-BB (100-14B: Peprotech). For phosphorylation studies by immunoblotting, the pericytes were lysed 30 min after PDGF-BB stimulation while cells were fixed 5 h later to check FoxO3 nuclear localization. Pre-treatment with cGKI inhibitor Rp-8-Br-PET-CGMPS (P007: Biolog) was carried out for 20 min before CNP stimulation, wherever applicable.

### Transfection with siRNA
Human pericytes were transfected with different siRNAs using Lipofectamine 3000 Transfection Reagent (Invitrogen) in an optiMEM serum-free

medium. As a control, commercially available non-targeting siRNA (si-Control) was used. All siRNA sequences are provided in Supplementary Table 1. 24 h after transfection, cells were cultured in a serum-containing medium for a resting period of 24 h, followed by stimulations for western blotting or BrdU incorporation assay.

## Subcellular fractionation/Immunoblotting

Cultured pericytes were lysed with a RIPA lysis buffer (Thermo-Scientific) containing protease and phosphatase inhibitors for whole cell lysis. For membrane fractionation, a subcellular fractionation kit (Nanotools) was used according to the manufacturer's instructions. SDS–PAGE and immunoblotting were performed as described previously[13]. The primary antibodies used are provided in Supplementary Table 2. GAPDH and Na$^+$–K$^+$ ATPase were used as loading control for whole cell lysates and membrane fractions, respectively. Protein bands were visualized with enhanced chemiluminescence and quantified by densitometry. All uncropped western blots are provided in Supplementary Fig. 4.

## Intracellular cGMP measurement

Cultured pericytes/PASMCs/MLVECs were serum starved for 3 h, followed by pre-treatment with phosphodiesterase (PDE) inhibitor 3-isobutyl-1-methylxanthine (0.5 mmol/L; Sigma) for 15 min. Thereafter, the cells were incubated with CNP, ANP (Bachem), or vehicle (saline) for an additional 15 min. Intracellular cGMP was extracted with ice-cold 70% ethanol and determined by radioimmunoassay[13].

## FoxO3 nuclear localization by immunocytochemistry

Human pericytes cultured on glass coverslips in 24-well plates were treated and fixed with acetone–methanol (1:1), washed 3 times for 5 min with PBS and blocked for 1 h with blocking buffer (5% BSA, 0.5% goat serum, 0.2% Triton-X in PBS), and incubated overnight with a FoxO3 primary antibody (1:200). This was followed by 1 h incubation with secondary antibody Alexa Fluor®-488 (1:1000, Life Technologies, A11008). After incubation, coverslips were dried and mounted on glass slides with DAPI containing fluorescent mounting medium. Fluorescent images were taken with an Olympus X microscope. The nuclear staining was quantified by Image J and plotted as a fold change of PDGF-BB-stimulated cells.

## Assessment of migration with gap closure assay

Pericyte migration was assessed using ibidi two-well culture inserts under different conditions. Briefly, ibidi two-well culture inserts were attached to each well of a 12-well plate, and 20,000 pericytes were seeded in each well. The pericytes were allowed to reach confluence and then subjected to serum starvation for 16 h. The culture insert was subsequently removed, and the cells were stimulated with either 10 or 100 nM of CNP. After a 30-min incubation, the cells were further stimulated with 30 ng/ml of PDGF-BB. Unstimulated cells were kept as controls. Immediately following the addition of PDGF-BB, the initial images of the gaps were captured and labeled as the zero-time point. The location of the gaps was marked to ensure consistent imaging for subsequent time points. Additional images were captured at 4, 8, 16, 20, and 24 h, maintaining the same position in each well. The quantification of gap closure was performed by measuring the gap size at the zero-time point and calculating the percentage of closure at each subsequent time point using Image J software.

## Assessment of proliferation

The influence of different treatments on proliferation was assessed with BrdU incorporation assay (Roche Diagnostics) according to the manufacturer's instructions. Briefly, 22 h after stimulation, cells were incubated with BrdU labeling solution for 2 h. After incubation, cells were fixed for 30 min, incubated with anti-BrdU peroxidase antibody for 90 min and finally washed with saline and incubated with substrate solution until color development. Absorbance was measured at 370 nm with reference at 492 nm in a plate reader (TECAN, Germany). The proliferation of cells was plotted as a fold change of absorbance compared to the PDGF-BB-stimulated cell's absorbance.

## mRNA expression analysis

cDNA from controls/PAH lungs and controls/MCT rat lungs were provided by Dr. Novoyatleva and Prof. Schermuly[27]. cDNA samples from NOX/HOX mice lungs were used from the previous publication[28]. Real-time RT-PCR was performed using a LightCycler Instrument (Roche). The quantitative data were calculated from the kinetic curve of the PCR by interpolation with a standard curve generated using known amounts of the target DNA.

The sequences of the primers are given in Supplementary Table 3.

## Statistics and reproducibility

Statistical analysis was performed with GraphPad Prism Software. All data sets are presented as means ± SEM. The individual statistical tests and sample sizes for each set of data are provided in the legends of figures and tables.

Data were tested for normality (Shapiro–Wilk test). For normally distributed data, Student's unpaired $t$-test for comparing two groups and ordinary one-way ANOVA for multiple comparisons with Tukey's post hoc test were employed. For grouped data, two-way ANOVA was used. For data that were not normally distributed, the nonparametric Mann–Whitney $U$ test was used for a 2-group comparison, and the Kruskal–Wallis analysis was performed for multiple groups. The difference with $p < 0.05$ between the groups was considered significant.

## Data availability

Numerical source data for all figures can be found in Supplementary Data 1. All other data that support the findings of this study are available from the corresponding author (S.D.) upon reasonable request.

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

## Acknowledgements

This work was supported by the Deutsche Forschungsgemeinschaft (DA 2462/1-1 to S.D.; KU 1037/8-1, KU 1037/12-1 and CRC1525 (project number 453989101) to M.K.) and by Else Kröner Fresenius Stiftung (2020_EKEA.131 to S.D.).

## Author contributions

S.D. developed the study concept, designed, and conducted experiments, analyzed the data, wrote the first draft of the manuscript, revised subsequent drafts, and took care of funding. M.N. isolated lung pericytes from control lungs, performed experiments, analyzed data, and generated figures. F.W., S.H., T.N., L.K., and K.V. performed experiments and/or analyzed the results. C.M. and I.A. provided lung biopsies for the isolation of lung pericytes. V.A.d.J.P. provided lung pericytes isolated from control and PAH lungs. T.N. and R.T.S. provided samples for mRNA expression analysis. M.K. developed the study concept, interpreted the results, reviewed and revised the manuscript, and took care of funding. All authors have approved the final version of the manuscript.

## Funding

## Competing interests

The authors declare no competing interests.
