## [Peer Review File · Communications Biology]

Reviewers' comments:

Reviewer #1 (Remarks to the Author):

This study aimed to investigate the potential impact of CNP/GC-B/cGMP/FoxO3 signaling on the functional and molecular changes observed in lung pericytes from patients with PAH. Dabral et al. effectively demonstrated the influence of CNP/GC-B and its downstream cGMP/cGKI signaling on pericyte behaviors, including proliferation and migration triggered by PDGF-BB, as well as the dedifferentiation process induced by TGF-beta 1. These findings enhance the current comprehension of the role that pericytes play in the remodeling of pulmonary vasculature associated to PAH. However, the robustness of the data is compromised by the limited number of biological samples, which undermines the strength of the drawn conclusions. Additionally, the reliance exclusively on in vitro observations poses a further limitation.

MAJOR concerns

1- Strengthening the credibility of these findings necessitates further steps. In vivo investigations employing animal models of pulmonary hypertension, such as monocrotaline or sugen/hypoxic rats, are not only recommended but essential. These investigations should be complemented by the integration of immunostaining techniques to authenticate the significance of key findings, such as the activation of PTEN, AKT, and Foxo3.

2- The restricted use of biological replicates within this study presents a notable drawback, impeding rigorous statistical analyses and subsequently weakening the solidity of the conclusions drawn. Observable disparities within the control group (Fig1C, Fig3B) and the utilization of only n=2 for experiments featured in Fig.1b, Fig.3b, and Fig. 5C further emphasizes this limitation.

3- It is imperative to determine the expression levels of PDGF and TGF-beta receptors, given their well-established overexpression in PAH cells.

4- To enhance the relevance of the study, evaluating CNP levels in both PAH patients and animal models is recommended.

5- It is advisable to incorporate individual data points in all graphs to provide a clearer and more precise visualization of the results.

6- The study would significantly benefit from a more comprehensive characterization of the pericyte cultures used for experimentation. This can be achieved by exploring a panel of specific markers such as NG2, PDGFR-beta, CD146, nestin, as well as examining other cell types like α -SMA, CD31, CD160.

7- Clinical and demographic data related to patients with PAH are missing.

Reviewer #2 (Remarks to the Author):

Comments for the Authors:

In this manuscript, titled "C-type natriuretic peptide/cGMP/FoxO3 signaling attenuates hyperproliferation of pericytes from patients with Pulmonary Hypertension", Swati Dabral and colleagues elucidate that CNP's anti-proliferative effects are mediated through the cGMP-dependent protein kinase I and the inhibition of the PI3K/AKT pathway. This subsequently results in the nuclear stabilization and activation of the FoxO3 transcription factor. Furthermore, it's posited that CNP/GC-B/cGMP signaling in pericytes counters PDGFBB-induced proliferation and migration. However, the experimental framework lacks depth, necessitating further evidence to enhance the validity and reliability of the experimental data.

Major Comments:

1. Certain Western Blot results depicted in the manuscript, specifically in Fig 1C, Fig 3C, and Fig 6A (pFoxO3Thr32), appear blurred. I recommend the authors revisit and improve the clarity of these images.
2. It would be better for the authors to discern and report the statistical significance of differences between specific group comparisons to validate their conclusions. For instance, in Fig 6A, each group pre and post- the addition of Rp-8-Br-PET-cGMPs should be meticulously compared. Similar scrutiny is required for Fig 5D and Fig 8B.
3. It's imperative to incorporate either biological or technical replicates in experiments depicted in Fig 1C and Fig 6A. This will mitigate the influence of outliers and enhance the precision of the findings.
4. The manuscript would benefit from more robust evidence to characterize cell proliferation. Consider introducing IF or WB assays for PCNA protein and employing fluorescent staining or flow cytometry for EdU.

Minor Comments:

1. The line plots in Fig 2B and 2F are missing dotted lines. Additionally, the significance markers are absent in the bar graph of Fig 3B.
2. The manuscript contains typographical errors, such as "supressing" (line 236) and "mimicks" in the title of Figure 7. Kindly address these for clarity.

Our answers to Reviewers' comments:

Reviewer #1 (Remarks to the Author):

We would like to thank the reviewer very much for his/her interest in our study and the appreciation that our results “effectively demonstrated the influence of CNP/GC-B and its downstream cGMP/cGKI signaling on pericyte behaviors”. His/her comment that “our findings enhance the current comprehension of the role that pericytes play in the remodeling of pulmonary vasculature associated to PAH” was very motivating for us. We performed additional experiments to address all major and specific concerns and feel that our manuscript has markedly improved.

Reviewers comments are listed below, followed by our answers (in blue) and an indication where they were addressed in the manuscript (underlined).

MAJOR concerns

#1 Comment 1. Strengthening the credibility of these findings necessitates further steps. In vivo investigations employing animal models of pulmonary hypertension, such as monocrotaline or sugen/hypoxic rats, are not only recommended but essential.

Response 1. To follow-up the *in vivo* relevance of our findings, we investigated the mRNA expression of CNP and its receptor, GC-B, in lung samples from two animal models of pulmonary hypertension: monocrotaline (MCT)-induced PAH in rats and chronic hypoxia (HOX)-induced PH in mice. In both preclinical models the pulmonary expression levels of CNP were markedly reduced. Interestingly, GC-B levels were unaltered in MCT rats while significantly decreased in HOX mice (HOX). Most importantly, in line with the experimental models, CNP expression levels were also significantly reduced in lung samples from patients with PAH as compared to control lungs, whereas GC-B levels were unaltered.

These new data add relevant information to our studies of cultured human lung pericytes. Since exogenous synthetic CNP prevented their growth factor induced alterations, it is possible that the endogenous paracrine-acting endothelial peptide physiologically “stabilizes” adjacent pericytes and that such effects are impaired in pulmonary hypertension. This could contribute to the progression of vascular remodeling in this disease. Since the expression and/or activity of the GC-B receptor was preserved in pericytes and in lung tissue samples from patients with PAH, substitution of the endogenous hormone by exogenous synthetic peptide may have protective vascular effects, as already suggested by preclinical studies (see refs. 14 and 15 in the manuscript).

Please see results section (page 9 lines 267 – 284) and corresponding new Figure 8; and discussion (on page 13 -14).

#1 Comment 2. These investigations should be complemented by the integration of immunostaining techniques to authenticate the significance of key findings, such as the activation of PTEN, AKT, and Foxo3.

Response 2. To this comment we performed co-stainings of phospho-AKT (pAKT_{Ser473}), FoxO3 and PTEN with the pericyte marker NG2 in lung sections from MCT- vs vehicle-treated rats.

The immunoreactive signal for pAKT_{Ser473} was markedly increased in lungs from MCT rats (see Figure R1 below, for review purposes only). Although multiple types of cells were positive for phospho-AKT, a strong co-localization was detected with NG2, indicating that AKT is activated in lung pericytes under the disease conditions of MCT-induced PH. PI3K/AKT signaling is a ubiquitous intracellular signaling pathway promoting cell growth. It was shown that its activation in pulmonary arterial smooth muscle cells (PASMC) and endothelial cells contributes to vascular remodeling in PH (Garat *et al.*, *J Cardiovasc Pharmacol* **62**, 539-548 (2013); Shi *et al.*, *Hypertension* **80**, 1035-1047 (2023)). Our immunofluorescence stainings corroborate these previous findings and demonstrate that this pathway is also activated in pericytes *in situ*.

Figure R1: Immunohistochemistry shows pAKT_{Ser473} co-localization with NG2 (pericyte marker) in lungs of MCT- and vehicle-treated rats. Please note that the signal for pAKT_{Ser473} is overall increased in the MCT lungs, including pericytes.

pAKT_{Ser473} (Mus; proteintech: 66444-1-Ig; 1:200) was co-stained with NG2 (Rb; Sigma: ab5320; 1:300) in lung cryo-sections from MCT and vehicle treated rats. Images were acquired with Leica SP8 confocal microscope and processed with Image J. Each image is representative from one animal. DAPI was used as a nuclear stain. Scale bar: 200 μ m.

FoxO3 positive staining was observed in vehicle-treated rats and partly colocalized with the NG2 signal (see Figure R2 below). FoxO3 immunoreactivity was strongly reduced in lungs of MCT-treated rats confirming reduction of FoxO3 expression in pulmonary hypertension.

Figure R2: Immunohistochemistry shows FoxO3 co-localization with NG2 (pericyte marker) in lungs of MCT- and vehicle-treated rats. Please note that the signal for FoxO3 is overall decreased in the MCT lungs, including pericytes.

FoxO3 (Mus; SantaCruz: sc-48348; 1:200) was co-stained with NG2 (Rb; Sigma: zrb5320; 1:100) in lung cryo-sections from MCT and vehicle treated rats. Images were acquired with Leica SP8 confocal microscope and processed with Image J. Each image is representative from one animal. DAPI was used as a nuclear stain. Scale bar: 200 μ m.

Unfortunately, several anti-PTEN antibodies which we used in the immunofluorescence stainings did not exhibit the expected specificity required for reliable detection of PTEN (SantaCruz: sc-7974; abcam: ab267787). Despite our efforts to troubleshoot and optimize the experimental conditions, we did not obtain a specific positive signal for PTEN in lung sections.

These immunohistochemical studies corroborate the *in vivo* relevance of our *in vitro* findings, showing the activation of AKT and inhibition of FoxO3 in pericytes of hypertensive lungs *in situ*. Unfortunately, it is currently not possible for us to study whether infusion of CNP will inhibit these changes in MCT-treated rats. Currently in Germany it takes 9-12 months to obtain permissions for new animal studies, a time frame which is beyond the revision of our manuscript. Due to this limitation, we prefer to include these immunohistochemical data in the letter to the reviewer only, and not in the manuscript.

#1 Comment 3. The restricted use of biological replicates within this study presents a notable drawback, impeding rigorous statistical analyses and subsequently weakening the solidity of the conclusions drawn. Observable disparities within the control group (Fig 1C, Fig 3B) and the utilization of only n=2 for experiments featured in Fig.1b, Fig.3b, and Fig. 5C further emphasizes this limitation.

Response 3. We fully understand this concern. We performed additional experiments to enhance the biological replicates for control and PAH pericytes in the revised Figures 1C, 3B and 5C. The results corroborated our previous findings.

In Fig. 1B, there was a mistake in the legend of the figure. It was stated 'n = 8 from 2 biological replicates' while it was from 'n = 8 from 4 biological replicates. We apologize for this mistake. The legend has been corrected in the revised manuscript.

#1 Comment 4. It is imperative to determine the expression levels of PDGF and TGF-beta receptors, given their well-established overexpression in PAH cells.

Response 4. We performed new experiments to compare the expression of PDGF and TGF- β receptors between control and PAH pericytes by immunoblotting (n = 5 in each group). The results demonstrate a significant 2-fold elevation in PDGFR- β levels in PAH pericytes, indicating that their augmented responses to PDGF-BB are related at least in part to increased receptor expression. Additionally, TGF- β RII expression was slightly upregulated in PAH pericytes (P = 0.0604).

Please see results section (page 5 lines 127 - 130, page 6 line 159 - 160) and corresponding new Supplementary Figure 1; and discussion on page 11).

#1 Comment 5. To enhance the relevance of the study, evaluating CNP levels in both PAH patients and animal models is recommended.

Response 5 (please also see Response 1). CNP is rapidly cleared from the circulation and present at very low levels in plasma (*Stingo et al. Am J Physiol* **263**, H1318-1321 (1992)). Given the paracrine nature of CNP actions, the circulating CNP concentrations possibly do not reflect its pulmonary contributions, i.e. the local levels at the endothelial-pericyte interphase. Therefore, we investigated the mRNA expression of CNP and its receptor, GC-B, in lung samples from two animal models of pulmonary hypertension: monocrotaline (MCT)-induced PAH in rats and chronic hypoxia (HOX)-induced PH in mice. In both preclinical models the pulmonary expression levels of CNP were markedly reduced, while GC-B levels were unaltered (MCT) or mildly decreased (HOX). In line with this, CNP expression levels were also significantly reduced in lung samples from patients with PAH as compared to control lungs, whereas GC-B levels were unaltered.

These data suggest downregulation of protective endogenous CNP/GC-B signaling in pulmonary hypertension.

Please see results section (page 9 lines 267 – 284) and corresponding new Figure 8; and discussion (on page 13 -14).

#1 Comment 6. It is advisable to incorporate individual data points in all graphs to provide a clearer and more precise visualization of the results.

Response 6. The graphs were revised accordingly and data are presented as individual data points in all the figures.

#1 Comment 7. The study would significantly benefit from a more comprehensive characterization of the pericyte cultures used for experimentation. This can be achieved by exploring a panel of specific markers such as NG2, PDGFR-beta, CD146, nesting, as well as examining other cell types like α -SMA, CD31, CD160.

Response 7. The control (n=3) and PAH (n=5) pericytes were obtained from Prof. Vinicio A. de Jesus Perez, Stanford University, Stanford. The isolation method and the detailed characterization of the pericytes used in the current manuscript were described before (see *Reference 45 in the manuscript*). The culture protocols are described in the materials and methods section of our manuscript, while the characterization of the cells by FACS was described in Figure 1 of the manuscript previously published by Yuan *et al* (Reference 45 of the manuscript).

Accordingly, the following statement is provided in the Methods section of our revised manuscript:

'Human lung pericytes from control (n =3) and PAH lungs (n =5) were provided by Prof. Perez. The details of the isolation protocol and characterization were provided in the previous publication'.

Please see methods section (page 15 lines 453 456) and corresponding reference 45.

In addition, to increase the number of experiments performed with control pericytes, we isolated and cultured pericytes from lung samples obtained through the Department of Thoracic and Cardiovascular Surgery, Julius Maximilians University Würzburg. These were healthy lung tissues which were resected for therapeutic purposes in patients with lung cancer. The ethics approval and details of isolation, culture and characterization are provided in the methods section of the revised manuscript.

Please see methods section (page 15 lines 458 - 478).

#1 Comment 8. Clinical and demographic data related to patients with PAH are missing.

Response 8. The clinical data of these patients were provided in the Table 1 of a previous publication from Prof. Perez (please see reference 45 in the manuscript).

Please see methods section (page 15 lines 449 - 451) and corresponding reference 45.

Reviewer #2 (Remarks to the Author):

We would like to thank the reviewer very much for his/her interest in our study. We performed additional experiments to address all major and specific concerns and feel that our manuscript has markedly improved.

Reviewers comments are listed below, followed by our answers (in blue) and an indication where they were addressed in the manuscript (underlined).

Major Comments:

#2 Comment 1. Certain Western Blot results depicted in the manuscript, specifically in Fig 1C, Fig 3C, and Fig 6A (pFoxO3Thr32), appear blurred. I recommend the authors revisit and improve the clarity of these images.

Response 1. The representative images for these western blots have been replaced. Please see revised Figures 1C, 3C and 6A.

#2 Comment 2. It would be better for the authors to discern and report the statistical significance of differences between specific group comparisons to validate their conclusions. For instance, in Fig 6A, each group pre and post- the addition of Rp-8-Br-PET-cGMPs should be meticulously compared. Similar scrutiny is required for Fig 5D and Fig 8B.

Response 2. For clarity, the statistical method used for each data set is now provided in the corresponding figure legend.

Concretely:

- The 2-way ANOVA, followed by Tukey's multiple comparison test, was employed to assess the statistical significance of differences across multiple groups, specifically in Figures 6A, 6C, 8B (Fig 7B in revised manuscript) and Suppl. Figure 2.
- The data in Figure 6B are not normally distributed. Therefore, instead of 2-way ANOVA, the non-parametric Kruskal Wallis Test was employed.
- For data sets comparing treatments (e.g. PBS, PDGF ± CNP) after distinct pretreatments, the differences between treatment conditions and the influence of pretreatments were both statistically evaluated. The following symbols are used to depict the statistical differences (as mentioned in the figure legends): * $p < 0.05$ vs respective PBS (-), # $p < 0.05$ vs respective PDGF-BB, \$ $p < 0.05$ vs the corresponding vehicle-pretreated group. This specifically concerns Figure 6A (pretreatment: Vehicle or Rp-8-Br-PET-cGMPs) and Figure 5D (pretreatment: si-Control versus si-FoxO3 transfected).

#2 Comment 3. It's imperative to incorporate either biological or technical replicates in experiments depicted in Fig 1C and Fig 6A. This will mitigate the influence of outliers and enhance the precision of the findings.

Response 3. We fully understand this concern. We performed additional experiments to enhance the biological replicates for control and PAH pericytes in the revised Figures 1C, 3B, 5C and 6A. The results corroborated our previous findings.

In Fig. 1B, there was a mistake in the legend of the figure. It was stated 'n = 8 from 2 biological replicates' while it was from 'n = 8 from 4 biological replicates. We apologize for this mistake. The legend has been corrected in the revised manuscript.

#2 Comment 4. The manuscript would benefit from more robust evidence to characterize cell proliferation. Consider introducing IF or WB assays for PCNA protein and employing fluorescent staining or flow cytometry for EdU.

Response 4. To this comment, we performed new immunoblots for proliferating cell nuclear antigen (PCNA) protein. In line with the BrdU incorporation assay (Figure 2D) and cyclin D1 expression (Figure 2E), PDGF-BB (50 ng/ml, 24 h) led to a significant upregulation of PCNA expression in both control and PAH pericytes. CNP pretreatment (100 nM, 30 min) abolished this effect (new Figure 2F in the revised manuscript).

Please see results section (page 5 lines 147 - 150) and corresponding figure 2F.

We also studied whether the effect of CNP on PDGF-BB induced PCNA expression is cGKI dependent. For this, control pericytes were pretreated with the cGKI inhibitor (Rp-8-Br-PET-cGMPs: 100 μ M, 20 min) followed by CNP (30 min) and PDGF-BB treatment (50 ng/ml, 24 h). The cGKI inhibitor completely prevented the effect of CNP on PDGF-BB induced PCNA expression. These results, which corroborate our previous findings, are provided in the new Figure 6C of the revised manuscript.

Please see results section (page 8 lines 228 - 232) and corresponding figure 2F.

#2 Minor Comments:

#2 Comment 5. The line plots in Fig 2B and 2F are missing dotted lines. Additionally, the significance markers are absent in the bar graph of Fig 3B.

Response 5. The missing dotted lines have been added.

In figure 3B, the statistics test was not carried out as n =2 replicates were used. We performed additional experiments to enhance the biological replicates for control and PAH pericytes and perform statistical tests and the results are provided in the revised Figure 3B.

#2 Comment 6. The manuscript contains typographical errors, such as "supressing" (line 236) and "mimicks" in the title of Figure 7. Kindly address these for clarity.

Response 6. We thank the reviewer for careful reading. The typographical errors have been corrected.

REVIEWERS' COMMENTS:

Reviewer #1 (Remarks to the Author):

The authors have addressed my comments and concerns and I have no further comments.

Reviewer #2 (Remarks to the Author):

The authors have satisfactorily addressed all of my previous concerns. I have no further comments at this time.